# Knowledge Graph Enhanced Generative Multi-modal Models for Class-Incremental Learning

**Xusheng Cao**[2], **Haori Lu**[2], **Linlan Huang**[2], **Fei Yang**[1,2], **Xialei Liu**[1,2]*, **Ming-Ming Cheng**[1,2]

[1]NKIARI, Shenzhen Futian, [2]VCIP, CS, Nankai University

{caoxusheng, luhaori, huanglinlan}@mail.nankai.edu.cn,
{feiyang, xialei, cmm}@nankai.edu.cn

## Abstract

Continual learning in computer vision faces the critical challenge of catastrophic forgetting, where models struggle to retain prior knowledge while adapting to new tasks. Although recent studies have attempted to leverage the generalization capabilities of pre-trained models to mitigate overfitting on current tasks, models still tend to forget details of previously learned categories as tasks progress, leading to misclassification. To address these limitations, we introduce a novel Knowledge Graph Enhanced Generative Multi-modal model (KG-GMM) that builds an evolving knowledge graph throughout the learning process. Our approach utilizes relationships within the knowledge graph to augment the class labels and assigns different relations to similar categories to enhance model differentiation. During testing, we propose a Knowledge Graph Augmented Inference method that locates specific categories by analyzing relationships within the generated text, thereby reducing the loss of detailed information about old classes when learning new knowledge and alleviating forgetting. Experiments demonstrate that our method effectively leverages relational information to help the model correct mispredictions, achieving state-of-the-art results in both conventional CIL and few-shot CIL settings, confirming the efficacy of knowledge graphs at preserving knowledge in the continual learning scenarios.

## 1 Introduction

Continual learning without forgetting old knowledge[34] has been thriving in the ever-changing world. Traditional approaches in this area typically employ three categories [33] of methods to prevent forgetting: replay-based methods [25, 40, 15, 57] that retain a portion of past data, architecture-based methods [32, 42, 58] that progressively expand the model, and regularization-based methods [7, 24, 1] that prevent excessive changes in model parameters. Recently, methods based on large-scale pre-trained models [21, 39] have attracted more attention due to their superior performance than the traditional train-from-scratch methods. For example, prompt-based methods [44, 56, 55, 54] have been proposed that use a small number of parameters to learn "general" and "specific" knowledge without altering the pre-trained backbone. SLCA [62] proposes to use a small learning rate for the backbone to preserve the generalizability and a larger learning rate for the classifier to accommodate new classes. Yet, these methods do not leverage textual information of the class labels or the relationships between the learned classes.

To leverage the rich information lies in text modality, Continual CLIP [49] proposes to use Frozen CLIP to conduct predictions based on image-text similarities. RAPF [18] uses CLIP [39] text encoder to detect and pull apart similar classes. PROOF [67] proposes to use task-specific projections on both image and text encoder. MoE-Adapters [59] utilize the Mixture-of-Experts adapters on top of the

---

*Corresponding author.

pre-trained CLIP model and an Auto-Selector to route for different input. CLIP-CIL [29] proposes to fine-tune an additional adapter after the backbone to learn new classes. However, in the more challenging exemplar-free scenario, these methods still face problems with classification bias towards the newly learned knowledge and forgetting of previously learned knowledge.

In contrast, GMM [5] addresses the bias problem by discarding the classifier and directly using a Large Language Model (LLM) to generate predicted text. GMM employs image-text pairs as input and only fine-tunes a linear layer, effectively mitigating bias and leveraging the LLM's capability to understand and generate human-interpretable text. However, two challenges persist during the learning process.

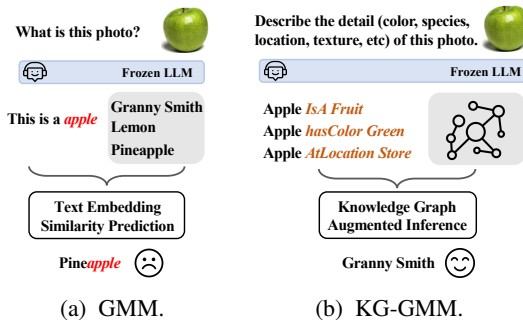

(a) GMM.          (b) KG-GMM.

Figure 1: The difference of inference pipeline between the generative-based baseline GMM [5] and our Knowledge Graph enhanced generative multi-modal model (KG-GMM).

Firstly, fine-tuning a small linear layer with a single, fixed format could cause the Multi-modal LLM to lose its generalization capabilities, causing it to output text only in one format "This is a photo of a [CLS]" while losing the ability to describe details about the image content in terms of colors, background, texture, etc. Secondly, since the model lacks exposure to images from previous tasks, it tends to classify all images encountered into higher-level categories (knowledge obtained in the pre-training phase). For example, if the model learns "Granny Smith" in the initial task, it can accurately output "Granny Smith" for images of that class during immediate testing. However, after learning other similar categories like "Pineapple" or "Lemon" in subsequent tasks without replaying exemplars, the model tends to respond, "This is a photo of an apple" when presented with green apple images during testing and misclassifying it to a newly learned text related label "Pineapple" as shown in Fig. 1.

To address these challenges, we turn to common sense knowledge graphs [19], which are structured collections of factual triples, organized as (head, relation, object), e.g., (Granny Smith, IsA, Fruit) that capture relationships between entities. Compared to the black-box generation process of LLMs [36], knowledge graphs encode rich interrelationships information in the form of human-readable language. For example, ConceptNet [45] is a general common sense knowledge graph integrating triplets that cover nearly all visual recognition datasets (including both coarse-grained and fine-grained ones) with efficient query operations. We believe that a knowledge graph that incrementally expands as tasks increase would be lightweight and straightforward while retaining the knowledge structures learned by the model to prevent forgetting.

Building on this idea, we propose a Knowledge Graph Enhanced Generative Multi-modal Model for class-incremental learning. It enables the model to focus more on the factual content within the images, providing descriptive output rather than direct guesses. Additionally, during inference, we construct a subgraph from the model's original textual output and compare it with the existing graph to identify the specific category to which the image belongs. Specifically, we use a common-sense knowledge graph to incrementally build a sub-graph during continual learning by storing class-relevant triplets. Training employs relations (not plain text) as ground truth labels. During testing, associative keywords (e.g., IsA, AtLocation) guide the model to output detailed image facts instead of direct guesses, enhancing precision by recalling prior relational knowledge from the graph. This enables structured retention and retrieval of learned relationships, improving answer specificity.

The main contributions of this paper are:

- We propose using an ever-expanding knowledge graph to help the model distinguish similar classes across different tasks based on relationships, providing more references when discriminating similar classes.

- During inference, we propose to guide the model with relation words to output factual descriptions rather than direct guesses, preventing forgetting through associative relationships.

- Experiments on multiple datasets and settings demonstrate that our method can help the model alleviate forgetting with minimal training cost.

## 2 Related Work

### 2.1 Class Incremental Learning

Class incremental learning is a method where a model learns new classes sequentially over time without forgetting the previously learned classes. Early methods of class-incremental learning often involved training from scratch [24, 11, 53, 69, 14, 60].

With the advent of pre-trained models, using them as a starting point for continual learning has shown great effectiveness. Initially, models pre-trained on ImageNet are typically fine-tuned using parameter-efficient methods to incorporate new knowledge, such as prompt-based techniques [41, 55, 56, 44] and low-rank adapters [13, 26]. Some approaches involve fine-tuning specific modules of the model [66] or setting differential learning rates [62]. Building on this, guidance from textual modality information further alleviates forgetting. The CLIP model, with its strong zero-shot capabilities, has significantly benefited class-incremental learning [49] and garnered considerable attention. This is often achieved by adding parameter modules, such as linear layers [29, 18] or attention modules [20, 67], to the original model to directly adjust the well-learned features.

Currently, generative multi-modal models demonstrate powerful capabilities at mitigating the forgetting problem in continual learning. Simple fine-tuning on these models can effectively retain old knowledge while learning new classes [5]. However, without replay, the GMM still tends to forget details of the classes learned in former tasks. Our method tackles this issue by incorporating rich relationship information in knowledge graphs to enhance the GMM learning and inference process.

### 2.2 Knowledge Graph

Knowledge graphs are structured representations of information where entities (nodes) are interconnected through relationships (edges), allowing for complex querying and inference over linked data. There are four main types of knowledge graphs [36]: Encyclopedic KGs that cover general knowledge (Wikidata [51]), Common Sense KGs (ConceptNet [45]) that capture everyday concepts and objects, Domain-specific KGs tailored to specialized fields (UMLS [3] for medical domain), and Multi-modal KGs [30] that integrate various data types such as text and images. We mainly focus on the Common Sense KG due to its broad scope that covers most classes in image datasets.

With the development of LLM, there has been research work that combines the generation capability of LLM models with the structured, rich factual knowledge stored in knowledge graphs. The combinations mainly fall into three categories: KG-enhanced LLMs [27, 38, 63] involve embedding KGs to improve LLMs by enhancing understanding and reasoning of the knowledge learned by LLMs. LLM-augmented KGs [4, 16, 23, 64] leverage LLMs for different KG-based tasks such as knowledge graph completion, graph-to-text generation, and question answering. The integration of LLMs and KGs [12, 22, 47] enables bidirectional reasoning grounded in both data and structured knowledge, as the two systems collaborate symbiotically to mutually enhance each other's capabilities.

While there have been studies on integrating knowledge graphs with LLMs, most of these focus on single-task scenarios. The potential of combining LLMs and knowledge graphs to enhance knowledge preservation in continual learning scenarios remains unexplored.

## 3 Method

In this section, we begin by introducing the setup of class-incremental learning and revisiting the training and testing procedures of GMM [5]. Then, we present our method from two perspectives: knowledge graph enhanced learning and knowledge graph augmented inference.

### 3.1 Preliminaries

**Class Incremental Learning (CIL).** CIL is a paradigm that focuses on the progressive acquisition of knowledge across disjoint sets of classes. Let $\mathcal{X}$ denote the input space and $\mathcal{Y}$ represent the label space. At each incremental time step $t$, the learning algorithm receives a new dataset $D_t = \{(x_i, y_i) \mid x_i \in \mathcal{X}_t, y_i \in \mathcal{Y}_t\}$, where $\mathcal{Y}_t$ is a set of novel classes such that $\mathcal{Y}_t \cap (\cup_{k=1}^{t-1} \mathcal{Y}_k) = \emptyset$. The objective is to construct a classifier $f_t : \mathcal{X} \to \bigcup_{k=1}^{t} \mathcal{Y}_k$ that predicts labels overall observed classes up to time $t$.

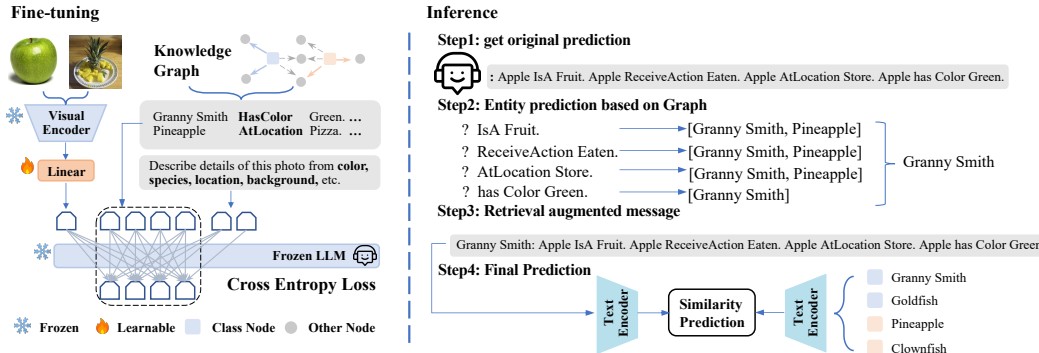

Figure 2: Left: In knowledge graph-enhanced learning, image embedding, knowledge-enhanced ground-truth embedding, and question embedding are input to Frozen LLM to generate predicted embedding; cross-entropy loss updates the linear layer. Right: In knowledge graph-augmented inference, relations from predicted text are extracted to create graph-augmented text, producing the final prediction.

**Generative Multi-modal Models for CIL.** In order to effectively harness the textual information embedded within the graph, we need to leverage language models capable of extracting text embeddings that can be aligned with image features. Currently, two continual learning frameworks can incorporate textual features: those based on CLIP [20, 29, 18] and those utilizing large language models (LLMs) [5]. Given our need for more nuanced text understanding and encoding-decoding abilities, we adhere to the continual learning framework outlined in GMM [5].

The GMM method's training pipeline involves using a frozen image encoder $f_{enc}$ and a text encoder $f_{text}$ to process the image-text pairs $\{(X_{t,i}, S_{t,i})\}_{i=1}^{N_t}$ for each task $t$, where $N_t$ means the total number of image-text pairs in task $t$ and $S_t$ is the text labels of classes in task $t$. For each image $x_i$, $f_{enc}$ generates an image embedding $\mathbf{e}_i = f_{enc}(x_i; \theta_{enc})$, and for each class, GMM utilize the BERT tokenizer to get the corresponding embedding $\mathbf{s}_i$. Then, a question embedding $\mathbf{q}$ is also computed by the BERT model from the question text $q$. The image embedding $\mathbf{e}_i$ concatenated with the ground-truth embedding $\mathbf{s}_i$ and question embedding $\mathbf{q}$, is input to a frozen LLM to get the predicted tokens sequentially based on prior tokens:

$$P(\hat{\mathbf{s}}_1, ..., \hat{\mathbf{s}}_m | x_i, \mathbf{q}, \mathbf{s}) = \prod_{j=1}^{m-1} P(\mathbf{s_j} | \mathbf{e}_i, \mathbf{q}, \mathbf{s_1}, \ldots, \mathbf{s_{j-1}}). \tag{1}$$

Then GMM computes Cross Entropy Loss to make the predicted token close to the GT token:

$$L_{CE} = -\frac{1}{m} \sum_{j=1}^{m} \mathbf{s}_j \cdot \log \hat{\mathbf{s}}_j, \tag{2}$$

where $\mathbf{s}_j$ is the ground truth and $\hat{\mathbf{s}}_j$ is the prediction.

At inference time, GMM uses the fine-tuned model to get the text prediction based on the test image and instruction question. Then a text encoder $f_{text}$ is used to measure the cosine similarity between predicted labels and ground truth to determine the final classification as:

$$\text{pred} = \arg\max \langle f_{text}(\mathbf{s}), f_{text}(\hat{\mathbf{s}}) \rangle. \tag{3}$$

As discussed in the introduction, GMM may misclassify a predicted text to its word-related class label instead of its meaning-related ground truth (e.g., apple to pineapple instead of apple to Granny Smith). So we propose to use relations in a common sense graph to instruct the model output relation-aware text, thus helping locate the predicted text to its ground truth. We begin by introducing the Knowledge graph construction process, followed by how to utilize this graph for training and inference.

### 3.2 Knowledge Graph Enhanced Learning

**Knowledge Graph Construction.** In order to extract relations for each encountered new category, we follow ZSL-KG [35] to query three tables: nodes, relations, and edges from the ConceptNet database based on the ILSVRC-21K [10] classes within two-hop relations.

Then, we have a common sense knowledge graph at hand presented as $G = \{E, R, F\}$, where $E$ represents entities, $R$ represents relations and $F = \{(h, r, o)\} \subseteq E \times R \times E$ represents facts contained in this graph, in which $h$ means head entity and $o$ means tail object. In the continual learning process, we gradually build a dataset-specific graph in the order in which classes appear at each given task.

To better illustrate our method, we divide our graph construction process into two steps:

**Step 1: Build a temporary knowledge graph.** Suppose we are currently in task $t$ of continual learning. We initialize

$$E_t = E_{t-1} \cup S_t \cup \{e \mid e \in \mathcal{T}(S_t)\},\tag{4}$$

where $S_t$ denotes new class nodes, and $\mathcal{T}(S_t)$ extracts non-class nodes from relation triplets involving $S_t$. Then we build a temporary graph for task $t$:

$$G_t = (E_t, R_t, F_t),\tag{5}$$

containing all entities, relations, and fact triplets observed up to task $t$.

**Step 2: Trim to Key Relationships.** To enable efficient training and improved discrimination between similar classes, we proposed to trim the temporary knowledge graph $G_t$ to a smaller one $G'_t$:

$$G'_t = (E'_t, R'_t, F'_t),\tag{6}$$

that only contains a few unique relations that best distinguish it from similar old classes. Here, $E'_t$ is the trimmed entity set:

$$E'_t = \left\{ e \mid e \in E_t, \ e \notin \bigcup_{k=1}^{t-1} E'_k \right\}.\tag{7}$$

$R'_t$ and $F'_t$ are relations and facts that are associated with the current entity set $G'_t$. For example, as illustrated in Fig. 3, we choose "AtLocation Store" and "AtLocation Pizza" instead of previously used "IsA Fruit" and "ReceiveAction Eaten" by class "Granny Smith" from task $t-1$. The whole construction process of $G'_t$ is detailed in Algo. 1 Lines 4-6.

There exist circumstances where classes from later tasks (maybe the last task) have no direct relation to use; that is, all directly related tail nodes have been taken by former encountered classes. We tackle this problem by using relations of the second hop in the common sense knowledge graph $G$ (e.g., we use "Clownfish ReatedTo Water RelatedTo River" instead of the previously occupied node "Water").

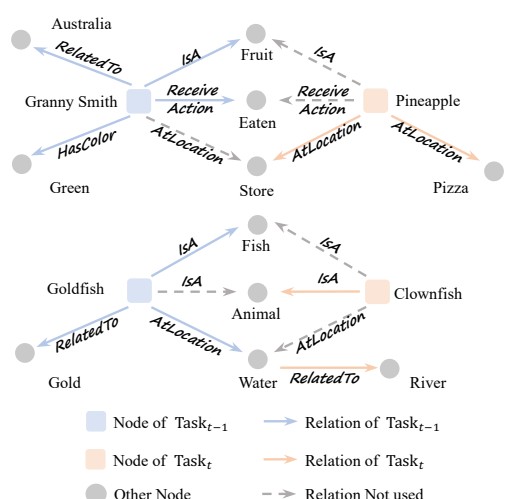

Figure 3: The knowledge graph construction process. Rectangle nodes represent class nodes, while round ones represent non-class nodes. Blue rectangles represent classes encountered in task $t-1$, while the orange ones represent classes from task $t$. Blue Arrows represent relations used for learning task $t-1$, while the orange ones represent relations used for task $t$.

After obtaining the distinct graph that has different relations for each class, we then concatenate $m$ triplets of each class together and input them into the model as the ground truth text $s_i$ in the image-text pairs as shown in the left side of Fig. 2, fine-tuning the linear layer after the image encoder with Cross Entropy loss in Eq. 2.

### 3.3 Knowledge Graph Augmented Inference

The inference process is illustrated on the right side of Fig. 2. We first get the relation-aware output by the fine-tuned LLM with instructing questions "Describe details of this photo from color, species, location, background, etc.". Then we decompose the raw output $s$ into $m$ triplets: $\{(h_p, r_p, o_p)\}_{p=1}^m$.

**Algorithm 1** Graph Construction For Task $t$

---

**Input:** $S_t$        ▷ Class names of task $t$
**require:** $G = \{E, R, F\}$        ▷ Entire Knowledge Graph
**require:** $E_{t-1}$        ▷ Entity node for task $t-1$
**require:** $G_{t-1}$        ▷ Subgraph for task $t-1$
  1:  $E_t = E_{t-1} \cup S_t \cup \{e \,|\, e \in \mathcal{T}(S_t)\}$        ▷ Init $E_t$
  2:  $G'_t = \{E_t, R_{t-1}, F_{t-1}\}$        ▷ Init subgraph for task $t$
  3:  **for** entity $e$ in $S_t$ **do**        ▷ Iterate each new classes
  4:     **for** $(h, r, o)$ in $F$ **do**        ▷ Find unused facts
  5:         **if** $h == e$ and $(h, r, o) \notin F_{t-1}$ **then**
  6:            $G'_t = \{E_t, R_t \cup r, F_t \cup (h, r, o)\}$
  7:  **return** $G'_t$        ▷ Return the subgraph for task $t$

---

After obtaining the predicted relations, we remove the head entities $h_p$ and use the relation pairs $\{(r_p, o_p)\}_{p=1}^m$ to search within the subgraph $G_t$ constructed for the current task. The head entity that appears most frequently in the search results is considered the predicted head entity:

$$\hat{h} = \arg\max_h \sum_{p=1}^m \mathbb{I}\left[(h, r_p, o_p) \in G_t\right], \tag{8}$$

where $\mathbb{I}[\cdot]$ is the indicator function that equals 1 if the triplet is correct ( $(h, r_p, o_p) \in E_t$) and 0 otherwise.

Although this graph-based prediction $\hat{h}$ can be treated as the final prediction, it neglects the rich information contained in the original raw text output $s$. To leverage both the knowledge reservation ability provided by the graph and the text understanding and reasoning ability provided by LLM, we prepend this graph-based prediction $\hat{h}$ to $s$ to obtain the graph-augmented output:

$$s_a = \hat{h} \oplus s, \tag{9}$$

where $\oplus$ denotes the concatenation operation, meaning that $s_a$ is formed by placing $\hat{h}$ directly before $s$. This augmented sentence $s_a$ is then input into the text encoder to perform similarity prediction between the encoded $s_a$ and all class features encountered so far to obtain the final prediction:

$$pred = \arg\max_{c \in S_t} \text{sim}\left(f_{text}(s_a), f_{text}(c)\right). \tag{10}$$

## 4 Experiments

### 4.1 Experiments Setup

**Datasets.** We test our model on two commonly used continual learning benchmarks: Tiny-ImageNet and ImageNet-R, and two few-shot continual learning benchmarks: CIFAR100 and Mini-ImageNet. We also conduct experiments on fine-grained and medical imaging datasets, and the corresponding settings and detailed results are provided in the supplementary material.

For conventional continual learning, we follow the two standard configurations used in GMM [5]: B0, in which all classes are equally divided among different tasks, and B100 (i.e. Tiny-ImageNet) in which the first task contains 100 classes (half of the dataset) and the rest are equally divided into subsequent tasks. For few-shot continual learning, we follow the data splits proposed by [48]. For both datasets, we divide the data into two parts: a base session and incremental sessions. The base session consists of 60 classes with full access to all associated data. Each incremental session follows a 5-way 5-shot setting, introducing 5 new classes with only 5 samples per class.

We build our expanding knowledge graph based on ConceptNet, which is a large-scale, multilingual knowledge graph that represents common-sense relationships between words and phrases in natural language. It consists of over 8 million nodes and approximately 21 million edges, connecting concepts through 50 relationships including "IsA", "PartOf", "UsedFor" and "HasProperty", etc.

**Implementation details.** For the Knowledge Graph part, we follow ZSL-KG [35] to use a 2-hop ImageNet-based knowledge graph extracted from ConceptNet [45] as the initial $G$ described in

Table 1: The performance comparison results between our method and other methods on Tiny-ImageNet and ImageNet-R.

| Type | Method | Exemplar | Tiny-ImageNet | | | | | | ImageNet-R |
| | | | B100-5 tasks | | B100-10 tasks | | B100-20 tasks | | B0-10 tasks |
| | | | Avg | Last | Avg | Last | Avg | Last | Last |
|------|--------|----------|-----|------|-----|------|-----|------|------|
| Conventional | EWC [24] | ✗ | 19.01 | 6.00 | 15.82 | 3.79 | 12.35 | 4.73 | 35.00 |
| | LwF [25] | ✗ | 22.31 | 7.34 | 17.34 | 4.73 | 12.48 | 4.26 | 38.50 |
| | iCaRL [40] | ✓ | 45.95 | 34.60 | 43.22 | 33.22 | 37.85 | 27.54 | - |
| | EEIL [6] | ✓ | 47.17 | 35.12 | 45.03 | 34.64 | 40.41 | 29.72 | - |
| | UCIR [15] | ✓ | 50.30 | 39.42 | 48.58 | 37.29 | 42.84 | 30.85 | - |
| | PASS [69] | ✗ | 49.54 | 41.64 | 47.19 | 39.27 | 42.01 | 32.93 | - |
| | DyTox [11] | ✓ | 55.58 | 47.23 | 52.26 | 42.79 | 46.18 | 36.21 | - |
| Discriminative PT models | Continual-CLIP[49] | ✗ | 70.49 | 66.43 | 70.55 | 66.43 | 70.51 | 66.43 | 72.00 |
| | L2P [56] | ✗ | 83.53 | 78.32 | 76.37 | 65.78 | 68.04 | 52.40 | 72.92 |
| | DualPrompt [55] | ✗ | 85.15 | 81.01 | 81.38 | 73.73 | 73.45 | 60.16 | 68.82 |
| | CODA-Prompt [44] | ✗ | 85.91 | 81.36 | 82.80 | 75.28 | 77.43 | 66.32 | 73.88 |
| | MoE-CLIP [59] | ✗ | 81.12 | 76.81 | 80.23 | 76.35 | 79.96 | 75.77 | 80.87 |
| | RAPF [17] | ✗ | 78.64 | 74.67 | 77.42 | 73.57 | 76.29 | 72.65 | 80.28 |
| | Linear Probe | ✗ | 74.38 | 65.40 | 69.73 | 58.31 | 60.14 | 49.72 | 45.17 |
| Generative PT models | Zero-shot | ✗ | 58.16 | 53.72 | 58.10 | 53.72 | 58.13 | 53.72 | 67.38 |
| | GMM [5] | ✗ | 83.42 | 76.98 | 82.49 | 76.51 | 81.70 | 76.03 | 80.72 |
| | KG-GMM (Ours) | ✗ | **86.17** | **81.86** | **84.37** | **78.16** | **83.18** | **78.32** | **84.29** |

section 3.2. $G$ contains 574270 nodes $E$, 50 relations $R$ and 1380131 edges $F$. For the Generative Multi-modal model part, we follow GMM to use the MiniGPT-4 [68] framework as the image and text encoder. In the B0 setting of all datasets, we employ a 200-iteration warmup with a learning rate of 3e-6 and a learning rate from 3e-5 to 3e-6 with a cosine decay scheduler in the following fine-tuning phase. In the B100 setting, we first employ a learning rate of 3e-6, and then on the subsequent tasks, we adopt a lower learning rate of 3e-7, both employing a cosine decay scheduler.

**Baselines and evaluating metrics.** We follow GMM [5] to compare with three different categories of methods, including conventional train-from-scratch based [40, 15, 65, 58, 11, 43, 25, 6, 69], discriminative pre-trained based [56, 55, 44] and generative pre-trained based [5]. The evaluation metrics for the experiments are defined as follows: "Avg" represents the model's average accuracy across all tasks, while "Last" denotes the model's accuracy on all test sets after fine-tuning the final task. For few-shot continual learning, we also add a new metric Harmonic Accuracy [37] (HAcc) to check the balanced performance between the base and new classes. $A_H = \frac{2 \times A_0 \times A_n}{A_0 + A_n}$, where $A_0$ is the acc of base classes and $A_n$ is the average of all classes. We report an average accuracy of three runs based on three different class orders. Please refer to the Supplementary Material for detailed results of different orders.

## 4.2 Experiments on Conventional CIL

In Tab. 1, we present experiments on 100 base classes with 5, 10, and 20 incremental tasks settings on Tiny-ImageNet, and B0-10 tasks setting on ImageNet-R. Our KG-GMM demonstrates superior performance across all settings of these two datasets. On Tiny-ImageNet, our method consistently outperforms others in all settings, particularly we outperform GMM by 2.93% on average in three settings in terms of Last accuracy. On ImageNet-R, our model achieved an impressive last task performance of 84.29%, surpassing the previous SOTA GMM by 3.57%, the previous best prompt-based method CODA-Prompt by 10.41%.

## 4.3 Experiments on Few-shot CIL

In Tab. 2, we present the comparison results of our method against other baselines on Mini-ImageNet for few-shot class incremental learning. The table shows that our model lags behind several ImageNet-21K pre-trained methods in the base session, it achieves state-of-the-art results in all subsequent eight sessions. Specifically, in the final session, our method reaches an accuracy of 78.07%, outperforming DualPrompt [55] by 21.26% points, CODA-Prompt [44] by 16.93% points, and the previous best GMM [5] by 2.89% points. Notably, our model leads GMM by 1.64% points in the first incremental

Table 2: Comparison results of our method with other conventional baselines and methods on the Mini-ImageNet and CIFAR100 for few-shot class incremental learning. The table includes one base task and eight incremental tasks. **PD** is the performance drop between the first and last session. * indicates our re-implementation based on PILOT [46].

| Method | Mini-ImageNet | | | | | CIFAR100 | | | | |
|---|---|---|---|---|---|---|---|---|---|---|
| | 0 | 4 | 8 | PD↓ | HAcc↑ | 0 | 4 | 8 | PD↓ | HAcc↑ |
| iCaRL | 61.31 | 30.49 | 17.21 | 44.10 | 32.45 | 64.10 | 27.93 | 13.73 | 50.37 | 41.45 |
| EEIL | 61.31 | 33.14 | 19.58 | 41.73 | 28.43 | 64.10 | 28.96 | 15.85 | 48.25 | 32.43 |
| LUCIR | 61.31 | 25.68 | 14.17 | 47.14 | 35.65 | 64.10 | 31.61 | 13.54 | 50.56 | 39.67 |
| TOPIC | 61.31 | 37.48 | 24.42 | 36.89 | 32.98 | 64.10 | 40.11 | 29.37 | 34.73 | 25.23 |
| CEC | 72.00 | 56.70 | 47.63 | 24.37 | 15.96 | 73.07 | 58.09 | 49.14 | 23.93 | 22.46 |
| F2M | 72.05 | 56.71 | 47.84 | 24.21 | 19.21 | 71.45 | 57.76 | 49.35 | 22.06 | 19.37 |
| MetaFSCIL | 72.04 | 57.58 | 49.19 | 22.85 | 14.35 | 74.50 | 59.48 | 49.97 | 24.53 | 3.60 |
| Entropy-reg | 71.84 | 57.01 | 48.21 | 23.63 | 19.29 | 74.40 | 59.71 | 50.14 | 24.26 | 11.53 |
| L2P* | 94.12 | 70.94 | 56.83 | 37.29 | 0.00 | 91.22 | 68.66 | 54.89 | 36.33 | 0.00 |
| DualPrompt* | 93.97 | 70.61 | 56.80 | 37.17 | 0.10 | 91.08 | 68.45 | 54.67 | 36.41 | 0.10 |
| CODA-Prompt* | **95.37** | 74.47 | 61.14 | 34.23 | 0.00 | **93.55** | 71.91 | 59.32 | 34.23 | 0.00 |
| Zero-shot | 58.08 | 58.19 | 54.95 | **3.13** | 52.47 | 74.13 | 72.59 | 67.93 | **6.20** | 64.75 |
| GMM | 89.35 | 83.61 | 75.18 | 14.17 | 71.45 | 91.53 | 85.65 | 81.47 | 10.06 | 75.43 |
| KG-GMM (Ours) | 90.99 | **85.63** | **78.07** | 12.93 | **74.81** | 91.83 | **86.23** | **82.37** | 9.46 | **79.68** |

session and extends this lead to 2.89 points in the final session, indicating that our approach more effectively mitigates forgetting through the integration of LLM and knowledge graphs.

Tab. 2 also presents the comparison results of our method against other baselines on CIFAR100 for few-shot class incremental learning. The results indicate that our method also achieves superior performance on low-resolution datasets, outperforming the previous state-of-the-art method GMM by 0.90% points,the traditional train-from-scratch method Entropy-reg [28] by 32.23% points, and the state-of-the-art prompt-based method CODA-Prompt [44] by 23.05% points in the final session. Note that although our method shows only a slight improvement over GMM in the last session, it achieves a 4.25% gain in Harmonic Accuracy. This indicates that our approach better balances learning new classes while retaining previously learned ones.

Table 3: Ablation study results of the description labels and the Graph-Augmented Inference on ImageNet-R B0-10 tasks.

| Method | Accuracy |
|---|---|
| GMM | 80.72 |
| GMM+Descriptions Labels | 80.95 |
| KG-GMM w/o Graph-Augmented Inference | 82.77 |
| KG-GMM | 84.29 |

Table 4: Time complexity analysis on the last task of B0-10 tasks setting of the ImageNet-R dataset.

| Relation Stored | r = 0 | r = 2 | r = 3 (Ours) | r=4 |
|---|---|---|---|---|
| Avg text length (c) | 30 | 51 | 75 | 102 |
| Generation cost (s) | 6.54 | 7.42 | 9.67 | 12.63 |
| Storage (MB) | 0 | 0.05 | 0.07 | 0.08 |
| Graph Inference (ms) | 0 | 0.46 | 0.53 | 0.62 |
| Accuracy (%) | 81.03 | 81.92 | 84.29 | 84.31 |

### 4.4 Further Analysis

**Ablation study on different components.** In Tab. 3, we present ablation studies to validate the effectiveness of our method. We first use the GPT-3.5 generated descriptions (using the same prompt question as our method) for each class as the text in the image-text pair of the original GMM. The results show that without the explicit guidance of the relation keywords, the rich information brought by the text has almost no improvement over the simple text labels used in the original GMM. Besides, we show that using only the head entity $\hat{h}$ provided by the search results in KG (third row in Tab. 3 shows considerable improvement (82.77%), but combined with the text output, our proposed KG-GMM achieves the best results 84.29%.

**Ablation on Max tokens.** In Fig. 5 we present the comparative analysis of maximum token limits on model performance and inference time. The experiments were conducted on ImageNet-R, comparing

| Image | Ground-Truth label | GMM predicted text | GMM prediction | KG-GMM predicted text | KG-GMM Prediction |
|---|---|---|---|---|---|
| | Granny Smith | This is a photo of a apple. | Pineapple | Apple IsA Fruit
Apple HasColor Green
Apple AtLocation Store | Granny Smith |
| | Hammerhead | This is a photo of a shark. | Great White Shark | Shark RelatedTo Head
Shark RelatedTo Hammer
Shark AtLocation Sea | Hammerhead |
| | Goldfinch | This is a photo of a bird. | Hummingbird | Bird RelatedTo Gold
Bird AtLocation Backyard
Bird IsA Finch | Goldfinch |
| | Fox Squirrel | This is a photo of a fox. | Red Fox | Fox IsA animal
Fox HasContext movies
Fox RelatedTo Red | Red Fox |
| | Afghan Hound | This is a photo of a horse. | Zebra | Horse RelatedTo barn
HorseSynonym Horse
Horse AtLocation farm yard | Barn |

Figure 4: Text examples of our methods against the original GMM.

our KG-GMM method with the baseline approach GMM+Descriptions that utilizes class descriptions generated by GPT-3.5 for data augmentation. Inference time was measured as the averaged processing duration per batch (size=64). Notably, our method achieves near-optimal performance at 20 tokens, while GMM+Descriptions exhibits slower performance growth with increasing tokens. This indicates that the knowledge graph-augmented GMM produces more informative outputs at lower token constraints, demonstrating two key advantages: 1) Enhanced information efficiency through KG-enhanced training enables effective knowledge condensation, and 2) Superior performance is attained with reduced computational overhead.

**Time complexity analysis.** In Tab. 4, we present the additional storage and inference costs associated with selecting different numbers of relations $r$ per class with a total batch size of 64. For $r = 0$, we used GMM's default prompt, "This is a photo of [CLS]", while for $r = 2$ and $r = 3$, the average generated text lengths correspond to 51 and 75 characters, respectively. "Generation cost" represents the time required to generate text, measured in seconds, "Storage" denotes the additional memory usage from the knowledge graph, measured in MB, and "Graph inference" indicates the time needed to obtain the graph-based prediction $\hat{h}$, measured in milliseconds.

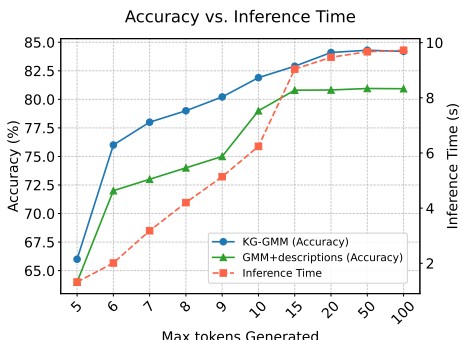

Figure 5: Model Accuracy vs. Inference Time comparison regarding max tokens configured during inference.

As shown in Tab. 4, the storage and inference overhead of the knowledge graph is minimal (0.07MB and 0.53ms for $r = 3$). The primary cost arises during the inference phase (generating longer texts increases inference time with higher values of $r$). Balancing inference time with performance gains, we select $r = 3$ as our final hyperparameter. We also conduct experiments on analysing memory overhead and training time between our method and the original GMM. Experimental results demonstrate that compared to the original GMM, our method requires a maximum memory footprint of 169 KB, while utilizing this additional memory allocation to preserve exemplars yields no corresponding performance improvement. We refer the readers to the Section A.3 for detailed experiments.

**Illustration of corrected predictions.** In Fig. 4 we present some visual examples of our proposed KG-GMM in terms of the predicted text and final prediction in the B0-10 tasks setting of the ImageNet-R dataset. We can see that the GMM can still recognize a shark but lose the ability to classify it to the right subclass (hammerhead). Instead, our KG-GMM can extract rich information from the relation contained in the predicted text and locate it to the right class. We refer the readers to the supplementary materials for more visualization results including failure cases.

# 5 Conclusions

In this paper, we propose KG-GMM, a novel method to combine MLLMs and knowledge graphs to tackle the catastrophic forgetting problem in continual learning. Our method gradually builds a graph in the process of learning new classes, assigning each class with $r$ different relations to enhance discriminability between semantic similar classes. During inference, we use the generated relations to locate the specific class, combined with the predicted text, our KG-GMM can effectively preserve much of the LLM's generalization ability while providing more accurate category predictions for given test images. Extensive experiments show that our method outperforms the state-of-the-art baselines for exemplar-free class incremental learning.

## Acknowledgments

This work is funded by NSFC (NO. 62206135, 62225604), Young Elite Scientists Sponsorship Program by CAST (2023QNRC001), the Fundamental Research Funds for the Central Universities (Nankai Universitiy, 070-63233085), Shenzhen Science and Technology Program (JCYJ20240813114237048), "Science and Technology Yongjiang 2035" key technology breakthrough plan project (2025Z053), Chinese government-guided local science and technology development fund projects (scientific and technological achievement transfer and transformation projects) (254Z0102G). Computation is supported by the Supercomputing Center of Nankai University.

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

# A  Appendix / supplemental material

## A.1  Different class order

Since the constructed sub-graph is influenced by the order in which classes are encountered, different class orders result in different sub-graphs. To evaluate this effect, we conducted multiple experiments with varying class sequences, and the results are presented in Tab. 5 and Tab. 6. As shown, while our model's performance fluctuates due to the order variations, it consistently maintains a high level of accuracy, demonstrating the robustness of our method.

Table 5: Comparison results between our method and other methods on Tiny-ImageNet and ImageNet-R under different class orders.

| Type | Method | Exemplar | Tiny-ImageNet | | | | | | ImageNet-R |
| | | | B100-5 tasks | | B100-10 tasks | | B100-20 tasks | | B0-10 tasks |
| | | | Avg | Last | Avg | Last | Avg | Last | Last |
|---|---|---|---|---|---|---|---|---|---|
| Conventional | EWC [24] | ✗ | 19.01 | 6.00 | 15.82 | 3.79 | 12.35 | 4.73 | 35.00 |
| | LwF [25] | ✗ | 22.31 | 7.34 | 17.34 | 4.73 | 12.48 | 4.26 | 38.50 |
| | iCaRL [40] | ✓ | 45.95 | 34.60 | 43.22 | 33.22 | 37.85 | 27.54 | - |
| | EEIL [6] | ✓ | 47.17 | 35.12 | 45.03 | 34.64 | 40.41 | 29.72 | - |
| | UCIR [15] | ✓ | 50.30 | 39.42 | 48.58 | 37.29 | 42.84 | 30.85 | - |
| | PASS [69] | ✗ | 49.54 | 41.64 | 47.19 | 39.27 | 42.01 | 32.93 | - |
| | DyTox [11] | ✓ | 55.58 | 47.23 | 52.26 | 42.79 | 46.18 | 36.21 | - |
| Discriminative PT models | Continual-CLIP[49] | ✗ | 70.49 | 66.43 | 70.55 | 66.43 | 70.51 | 66.43 | 72.00 |
| | L2P [56] | ✗ | 83.53 | 78.32 | 76.37 | 65.78 | 68.04 | 52.40 | 72.92 |
| | DualPrompt [55] | ✗ | 85.15 | 81.01 | 81.38 | 73.73 | 73.45 | 60.16 | 68.82 |
| | CODA-Prompt [44] | ✗ | 85.91 | 81.36 | 82.80 | 75.28 | 77.43 | 66.32 | 73.88 |
| | MoE-CLIP [59] | ✗ | 81.12 | 76.81 | 80.23 | 76.35 | 79.96 | 75.77 | 80.87 |
| | RAPF [17] | ✗ | 78.64 | 74.67 | 77.42 | 73.57 | 76.29 | 72.65 | 80.28 |
| | Linear Probe | ✗ | 74.38 | 65.40 | 69.73 | 58.31 | 60.14 | 49.72 | 45.17 |
| Generative PT models | Zero-shot | ✗ | 58.16 | 53.72 | 58.10 | 53.72 | 58.13 | 53.72 | 67.38 |
| | GMM [5] | ✗ | 83.42 | 76.98 | 82.49 | 76.51 | 81.70 | 76.03 | 80.72 |
| | KG-GMM (Ours) order1 | ✗ | 85.88 | 80.93 | 83.91 | 77.35 | 82.98 | 77.95 | 84.30 |
| | KG-GMM (Ours) order2 | ✗ | **86.35** | 82.02 | 84.45 | 78.47 | **83.43** | **78.66** | **84.62** |
| | KG-GMM (Ours) order3 | ✗ | 86.27 | **82.62** | **84.75** | **78.67** | 83.13 | 78.36 | 83.95 |
| | KG-GMM (Ours) | ✗ | 86.17 ± 0.21 | 81.86 ± 0.70 | 84.37 ± 0.35 | 78.16 ± 0.58 | 83.18 ± 0.19 | 78.32 ± 0.29 | 84.29 ± 0.27 |

Table 6: Comparison results of our method with other conventional baselines and methods on the mini-ImageNet dataset for few-shot class incremental learning under different class orders. The table includes one base task and eight incremental tasks. **PD** is the performance drop between the first and last session. * indicates our re-implementation based on PILOT [46].

| | 0 | 1 | 2 | 3 | 4 | 5 | 6 | 7 | 8 | PD↓ | HAcc↑ |
|---|---|---|---|---|---|---|---|---|---|---|---|
| iCaRL [40] | 61.31 | 46.32 | 42.94 | 37.63 | 30.49 | 24.00 | 20.89 | 18.80 | 17.21 | 44.10 | 32.45 |
| EEIL [6] | 61.31 | 46.58 | 44.00 | 37.29 | 33.14 | 27.12 | 24.10 | 21.57 | 19.58 | 41.73 | 28.43 |
| LUCIR [15] | 61.31 | 47.80 | 39.31 | 31.91 | 25.68 | 21.35 | 18.67 | 17.24 | 14.17 | 47.14 | 35.65 |
| TOPIC [48] | 61.31 | 50.09 | 45.17 | 41.16 | 37.48 | 35.52 | 32.19 | 29.46 | 24.42 | 36.89 | 32.98 |
| CEC [61] | 72.00 | 66.83 | 62.97 | 59.43 | 56.70 | 53.73 | 51.19 | 49.24 | 47.63 | 24.37 | 15.96 |
| F2M [43] | 72.05 | 67.47 | 63.16 | 59.70 | 56.71 | 53.77 | 51.11 | 49.21 | 47.84 | 24.21 | 19.23 |
| MetaFSCIL [8] | 72.04 | 67.94 | 63.77 | 60.29 | 57.58 | 55.16 | 52.90 | 50.79 | 49.19 | 22.85 | 14.35 |
| Entropy-reg [28] | 71.84 | 67.12 | 63.21 | 59.77 | 57.01 | 53.95 | 51.55 | 49.52 | 48.21 | 23.63 | 19.29 |
| L2P* [56] | 94.12 | 87.20 | 80.99 | 75.67 | 70.94 | 66.76 | 63.11 | 59.81 | 56.83 | 37.29 | 0.0 |
| DualPrompt* [55] | 93.97 | 86.85 | 80.67 | 75.31 | 70.61 | 66.44 | 62.77 | 59.58 | 56.80 | 37.17 | 0.1 |
| CODA-Prompt* [44] | 95.37 | 88.86 | 82.69 | 77.87 | 74.47 | 70.16 | 66.46 | 63.73 | 61.14 | 34.23 | 0.0 |
| Zero-shot | 58.08 | 58.95 | 57.76 | 57.89 | 58.19 | 57.42 | 56.26 | 54.82 | 54.95 | **3.13** | 52.47 |
| GMM | 89.35 | 88.40 | 86.11 | 85.07 | 83.61 | 81.35 | 78.97 | 77.34 | 75.18 | 14.17 | 71.45 |
| KG-GMM order 1 | 91.32 | 89.87 | 87.92 | 87.67 | 85.24 | 83.12 | 80.97 | 79.43 | 77.82 | 13.50 | 74.59 |
| KG-GMM order 2 | 91.54 | **90.14** | 88.13 | **87.83** | 85.42 | 83.69 | **81.45** | **80.23** | **78.43** | 13.11 | 74.87 |
| KG-GMM order 3 | 90.12 | 88.40 | **88.28** | 86.37 | **86.23** | **84.28** | 80.95 | 79.23 | 77.95 | 12.17 | **74.96** |
| KG-GMM avg | 90.99 ± 0.39 | 89.50 ± 0.61 | 88.11 ± 0.02 | 87.29 ± 0.43 | 85.63 ± 0.19 | 83.70 ± 0.22 | 81.12 ± 0.05 | 79.63 ± 0.19 | 78.07 ± 0.07 | 12.93 ± 0.31 | 74.81 ± 0.02 |

## A.2  Experiments on fine-grained and medical datasets

We conducted exploratory experiments on several fine-grained datasets and medical imaging datasets, as detailed below:

### A.2.1  Datasets and settings

CUB-200 [52] is a fine-grained bird species dataset with detailed part-level attributes. We construct a knowledge graph using each class's most certain attributes (confidence score 4), such as *has_bill_shape::hooked* or *has_bill_shape::cone*, resulting in approximately 17 relations per class after our graph construction method. All 200 classes are equally separated into 10 tasks.

HAM-10000 [50] is a medical image dataset of 10,000 pigmented skin lesions of 7 classes for melanoma classification. Since we could not find a suitable knowledge graph for medical data, we instead used DeepSeek-O1 to generate 10 representative attributes for each of the 7 classes (such as *color_dominance::light brown*, *background_skin::No pigmentation*), serving as their semantic descriptions or attribute sets. We follow [2] to separate 7 classes in 3 tasks, containing [2,2,3] classes each.

FVGC-Aircraft [31] is a dataset comprising 100 different aircraft classes. As no suitable external knowledge graph was available, we utilized the structured metadata provided with the dataset. Specifically, each aircraft class is described using three hierarchical levels: 41 manufacturers, 70 families, and 100 variants. These are represented as three relations—hasManufacturer, hasFamily, and hasVariant—with each class associated with exactly one relation at each level, resulting in three relations per class.

Herbarium [9] is a dataset containing 46,000 herbarium specimens across more than 680 species within the flowering plant family Melastomataceae. Due to the limited rebuttal time, it was difficult to process the entire dataset, so we randomly selected 10 species and used DeepSeek-O1 to generate six attribute-based relations: hasLeafShape, hasLeafMargin, hasLeafVeins, hasFlowersColor, hasStem, and hasFruit, each with corresponding class-specific properties. The selected classes were evenly divided into 5 tasks, with 2 species per task.

Table 7: Performance comparison across fine-grained and medical datasets.

| Datasets | CUB-200 | | HAM-10000 | | FGVC-Aircraft | | Herbarium19 | |
|---|---|---|---|---|---|---|---|---|
| | Avg | Last | Avg | Last | Avg | Last | Avg | Last |
| GMM | 49.34 | 40.39 | 79.22 | 63.81 | 51.63 | 47.25 | 86.34 | 79.88 |
| KG-GMM | 52.65 | 43.84 | 81.45 | 64.34 | 51.88 | 47.88 | 88.59 | 81.14 |

The experimental results in Tab. 7 demonstrate that, with sufficiently rich KG information (CUB-200, HAM-10000), our method consistently improves upon GMM. When relevant knowledge is limited (as in the case of FVGC-Aircraft), the performance gains from our method are relatively modest.

## A.3 Analysis on training overhead

In Tab. 8 we present more detailed comparisons on memory footprint (additional memory occupied by graph-based labels), training overhead (training time per batch), KG time (time used for constructing sub-graph $G'_t$), and the final accuracy. The experimental results demonstrate that the graph-based labels incur only a marginal overhead compared to word-based labels, requiring at most 169KB of additional memory and 0.07 seconds of extra processing time. Notably, when converting this 169KB capacity into exemplars for memory replay, the performance improvement proved statistically insignificant (from 80.72 to 80.91). This observation substantiates that our proposed method achieves substantial performance enhancements while maintaining minimal computational and storage overhead.

Table 8: Training overhead analysis on ImageNet-R.

| Method | Memory Usage | Training Time | KG Time | Acc |
|---|---|---|---|---|
| GMM | 244K | 0.36 | 0 | 80.72 |
| KG-GMM (task 1) | 332K | 0.40 | 2.46 | – |
| KG-GMM (task 6) | 368K | 0.41 | 2.57 | – |
| KG-GMM (task 10) | 413K | 0.43 | 2.73 | 84.29 |
| GMM + 3 exemplar | 484K | 0.36 | 0 | 80.91 |

## A.4 More visualization results

In Fig. 6, we provide additional visual examples of our methods against the original GMM. It can be observed that KG-GMM utilizes attributes such as the bird's color and location to refine its

| Image | Ground-Truth label | GMM predicted text | GMM prediction | KG-GMM predicted text | KG-GMM Prediction |
|---|---|---|---|---|---|
| | Granny Smith | This is a photo of a apple. | Pineapple | Apple IsA Fruit
Apple HasColor Green
Apple AtLocation Store | Granny Smith |
| | Great white shark | This is a photo of a shark. | Great white shark | Shark IsA Animal
Shark RelatedTo Great
Shark AtLocation Ocean | Great white shark |
| | Hammerhead | This is a photo of a shark. | Great white shark | Shark RelatedTo Head
Shark RelatedTo Hammer
Shark AtLocation Sea | Hammerhead |
| | Goldfinch | This is a photo of a bird. | Hummingbird | Bird RelatedTo Gold
Bird AtLocation Backyard
Bird IsA Finch | Goldfinch |
| | Tree frog | This is a photo of a toad. | Newt | Toad RelatedTo Frog
Toad AtLocation Tree
Toad IsA Amphibian | Tree frog |
| | Tarantula | This is a photo of a spider. | Spider web | Spider IsA Arachnid
Spider AtLocation USA
Spider RelatedTo Tarantula | Tarantula |
| | Cobra | This is a photo of a snake. | Iguana | Snake IsA reptile
Snake AtLocation zoo
Snake RelatedTo python | Iguana |
| | Basset Hound | This is a photo of a hound. | Afghan Hound | Hound IsA hound
Hound CapableOf bark
Hound IsA dog | Beagle |

Figure 6: More text examples of our methods against the original GMM. The first four examples illustrate how our method corrects the model drift in continual learning. The last two examples with red boxes, showcase where both our method and GMM made incorrect predictions.

classification to the finer-grained category of "goldfinch." In contrast, the original GMM, although retaining knowledge of the broader category, loses the ability to distinguish finer details after learning additional concepts. However, for certain categories that belong to the same broader class (e.g., hound) and have nearly identical characteristics (i.e., similar relations), our method may still make errors, though it typically misclassifies them into closely related categories (e.g., misclassifying a Basset Hound as a Beagle rather than an Afghan Hound based on text similarity). We believe that if similar classes were more distinctly separated with different relations or finer-grained graphs with detailed attributes were incorporated, it would further enhance our method's performance.

### A.5 Limitations and Border Impact

There are still several limitations in our work. As our work represents the first attempt to integrate knowledge graphs with continual learning, our experiments were primarily conducted on established continual learning benchmarks. We acknowledge that collecting KGs for some specialized datasets (e.g., fine-grained classification datasets and medical datasets) is tricky and remains unexplored. Furthermore, given that our primary baseline comparison focuses on GMM [5], our experiments were limited to consistent backbone architectures (EVA-CLIP and Vicuna-7B). Employing more advanced visual-language models could yield performance improvements. With the rapid development of LLMs, continual learning methods will become more important across various applications. Knowledge graphs emerge as a valuable yet previously underutilized tool for mitigating generalization loss during continual fine-tuning of LLMs. Exploring optimal methodologies for KG integration presents a promising research direction worthy of in-depth investigation.

