# OpenReview forum: "Knowledge Graph Enhanced Generative Multi-modal Models for Class-Incremental Learning"
_NeurIPS.cc/2025/Conference — NeurIPS 2025 poster_

### Official Review · Reviewer_qKfb · 2025-06-25

**Clarity:** 4
**Significance:** 3
**Originality:** 3
**Rating:** 4
**Confidence:** 5

**Summary:**

This paper proposes KG-GMM, which leverages a Knowledge Graph to enrich relational information for class incremental learning. It also introduces knowledge graph-augmented inference to preserve detailed knowledge and reduce forgetting. Experiments demonstrate its effectiveness in both class incremental and few-shot class incremental learning.

**Questions:**

Q1. While leveraging a knowledge graph to enrich relational information may help address the challenge of classifying at a higher semantic level, it remains unclear whether the proposed method adequately addresses the issue of losing generalization ability, because the proposed method still trains only simple linear layers.

Q2. As the knowledge graph is ImageNet-based, there is a concern of potential information leakage in the experiments conducted on ImageNet subsets (Tiny-ImageNet, ImageNet-R, Mini-ImageNet), since the model may have prior knowledge of both seen and unseen classes. Similarly, CIFAR-100 shares some class labels with ImageNet, raising the same concerns. It would strengthen the evaluation if the method were tested on fine-grained datasets, such as FGVC-Aircraft or Herbarium 19.

Q3. Which model is used for the "Zero-shot" performance at the experiments (Table 1 and Table 2)? Additionally, in Table 2, why is the performance drop observed in the zero-shot evaluation, which assesses the performance without training?

**Ethical Concerns:**

["NO or VERY MINOR ethics concerns only"]

**Final Justification:**

Thank you for the rebuttal. It has addressed all of my concerns, especially regarding the plasticity–stability dilemma in continual learning and the validation of the method’s general effectiveness. I have therefore decided to raise my score to borderline accept.

**Limitations:**

yes

**Paper Formatting Concerns:**

No concerns in this paper

**Quality:**

4

**Strengths And Weaknesses:**

## **Strengths**
1. The paper is well-written, with clear and visually appealing figures.

1. The authors provide a thorough background on Knowledge Graphs and Generative Multi-modal models for CIL, making the paper easy to follow.

## **Weaknesses**
1. It is uncertain whether the proposed method effectively addresses the two challenges of leveraging LLMs in CIL, as mentioned in the introduction.

1. The use of CIFAR-100 and subsets of ImageNet-21K is too limited to validate the general effectiveness of the proposed approach.

1. The method section lacks explanation on how each component contributes to class incremental learning, particularly for learning new classes or retaining old ones. As a result, the method appears to be merely a combination of well-pretrained models and a knowledge graph built from large datasets.

---

> ### Author Rebuttal · Authors · 2025-07-30
>
> Thank you for your valuable comments. We address your questions or concerns below.
> > **W1&Q1: Whether our method addresses the two challenges of leveraging LLMs in CIL.**
>
> Our method is designed to address two major challenges in using MLLMs for continual learning:
>
> - Fixed-format degeneration: Standard GMM setups often produce one-word responses due to training with only class label names. Our approach mitigates this by training with relational textual descriptions, which encourages the MLLM to retain its expressive capacity and generate more descriptive, attribute-rich outputs.
>
> - Semantic forgetting: MLLMs tend to regress to generic categories over time (e.g., predicting apple instead of Granny Smith). We address this by guiding inference with relational prompts and using attribute-based reasoning via a knowledge graph, enabling finer-grained class predictions even after multiple learning stages.
>
> Regarding the reviewer’s concern on generalization and the use of a simple linear layer:
> While we employ a lightweight linear classifier for efficiency and continual adaptation, it is important to note that the core generalization capacity lies in the frozen MLLM backbone, which is pretrained on massive and diverse image-text data. Our method does not aim to retrain or overfit a small model, but rather to prompt the MLLM to utilize its pretrained capabilities more effectively by supplying structured, discriminative knowledge. Moreover, our experiments (refer to **W2&Q2**) across diverse datasets (including fine-grained benchmarks like CUB-200 and HAM-10000) demonstrate that the method maintains good generalization and transferability.
>
> > **W2&Q2: About more experiments outside the ImageNet-21K scope.**
>
> We conducted exploratory experiments on several fine-grained datasets and medical imaging datasets, as detailed below:
> - CUB-200 is a fine-grained bird species dataset with detailed part-level attributes. We construct a knowledge graph using each class's most certain attributes (confidence score 4), such as has_bill_shape::hooked or has_bill_shape::cone, resulting in approximately 17 relations per class after our graph construction method. All 200 classes are equally separated into 10 tasks.
> - HAM-10000 is a medical image dataset of 10,000 pigmented skin lesions of 7 classes for melanoma classification. Since we could not find a suitable knowledge graph for medical data, we instead used DeepSeek-O1 to generate 10 representative attributes for each of the 7 classes (such as color_dominance::light brown, background_skin::No pigmentation), serving as their semantic descriptions or attribute sets. We follow [A] to separate 7 classes in 3 tasks, containing [2,2,3] classes each.
> - FVGC-Aircraft is a dataset comprising 100 different aircraft classes. As no suitable external knowledge graph was available, we utilized the structured metadata provided with the dataset. Specifically, each aircraft class is described using three hierarchical levels: 41 manufacturers, 70 families, and 100 variants. These are represented as three relations—hasManufacturer, hasFamily, and hasVariant—with each class associated with exactly one relation at each level, resulting in three relations per class.
> - Herbarium is a dataset containing 46,000 herbarium specimens across more than 680 species within the flowering plant family Melastomataceae. Due to the limited rebuttal time, it was difficult to process the entire dataset, so we randomly selected 10 species and used DeepSeek-O1 to generate six attribute-based relations: hasLeafShape, hasLeafMargin, hasLeafVeins, hasFlowersColor, hasStem, and hasFruit, each with corresponding class-specific properties. The selected classes were evenly divided into 5 tasks, with 2 species per task.
>
> [A] BiasPruner: Debiased Continual Learning for Medical Image Classification, MICCAI 2024
>
> | datasets | CUB-200 |      | HAM-10000 |      | FVGC-aircraft| | Herbarium 19| |
> |:--------:|:------:|:----:|:--------:|:----:|:----:|:----:|:----:|:----:|
> |          |   Avg  | Last |    Avg   | Last | Avg   | Last | Avg   | Last |
> |    GMM   | 49.34       | 40.39     |  79.22        |  63.81    | 51.63  | 47.25   | 86.34  | 79.88 |
> |  KG-GMM  |  52.65      |  43.84    |  81.45        |   64.34   |   51.88  |  47.88 |  88.59  |81.14    |
>
> The experimental results above demonstrate that, with sufficiently rich KG information (CUB-200, HAM-10000), our method consistently improves upon GMM. When relevant knowledge is limited (as in the case of FVGC-Aircraft), the performance gains from our method are relatively modest.
>
> >**W3: About the explanation on how our method addresses the plasticity and stability dilemma in continual learning.**
>
> Traditional CIL approaches mitigate forgetting by (1) storing exemplars, (2) expanding model architectures, or (3) constraining parameter updates. GMM takes a different route by reformulating classification as a generation problem, removing the static classifier and mitigating its bias toward new classes. However, as discussed earlier, this introduces two challenges: degeneration into fixed-format outputs and semantic drift toward generic categories.
>
> Our KG-GMM is designed to address these challenges and contributes to both retaining old knowledge (stability) and learning new classes (plasticity) in the following ways:
>
> - Relation-enriched descriptions during training help maintain the expressiveness of the MLLM and reduce degeneration, enhancing plasticity for new tasks.
>
> - Graph-augmented inference provides structured prompts that guide the model to recover fine-grained class details, directly supporting stability by retrieving specific attributes of previously learned classes.
>
> Empirically, our method shows stronger gains after multiple tasks are introduced, indicating enhanced knowledge retention. As shown in Table 2,  on Mini-ImageNet, improvement after Task 0 is modest (from 89.35 to 90.99), but becomes more evident by the final task (from 75.18 to 78.07). On CIFAR-100, gains grow from 91.53 to 91.83 (Task 0) and from 81.47 to 82.37 (final task).
> This performance trajectory confirms that KG-GMM’s components are not merely plug-ins, but actively contribute to mitigating forgetting and preserving discriminative knowledge across tasks.
>
> > **Q3: About "zero-shot" baseline and why it has a performance drop.**
>
> The baseline "zero-shot" follows the usage in the GMM paper, where MiniGPT-4 is directly prompted with the question "What is this a photo of?"  and the given test image. The model's textual response is then encoded using a CLIP text encoder and compared against all currently seen class embeddings—the most similar one is selected as the final predicted class. As continual learning progresses and the number of learned classes increases, the model is required to distinguish among more classes in a single step. As a result, even zero-shot models experience a noticeable drop in performance.

---

> > ### Comment · Reviewer_qKfb · 2025-08-05
> >
> > Thank you for the rebuttal. It has addressed all of my concerns, especially regarding the plasticity–stability dilemma in continual learning and the validation of the method’s general effectiveness. I have therefore decided to raise my score to borderline accept.

---

### Official Review · Reviewer_ErR5 · 2025-06-26

**Clarity:** 3
**Significance:** 3
**Originality:** 3
**Rating:** 5
**Confidence:** 5

**Summary:**

This paper addresses the challenge of class-incremental learning (CIL) by leveraging generative multimodal models (GMMs). To overcome the limitations of traditional GMMs, the paper integrates a knowledge graph into the generative framework. The incorporation of the knowledge graph enhances the retention of knowledge from old tasks, providing significant benefits in both the training and testing phases. Experimental results on standard CIL and few-shot CIL scenarios demonstrate substantial improvements over existing state-of-the-art methods.

**Questions:**

1. Could you clarify the main motivation behind the "Trim to Key Relationships" step? Is it feasible to use all possible relationships, or are there specific constraints guiding this choice?

2. Why does the "GMM + Description Labels" approach not perform as well, whereas GMM with a knowledge graph yields better results?

3. The "Inference Time" line in Fig. 5 is somewhat confusing. Could you provide further clarification on what this represents?

**Ethical Concerns:**

["NO or VERY MINOR ethics concerns only"]

**Final Justification:**

After reading the author's reply, I decided to keep my score.

**Limitations:**

Yes

**Quality:**

3

**Strengths And Weaknesses:**

Strengths:

1. The idea of integrating a knowledge graph into class-incremental learning (CIL) is both meaningful and novel. Current methods suffer from forgetting due to limited attention to knowledge relationships. This paper offers a simple yet effective solution by enhancing high-level graph knowledge with images.

2. The paper is well-structured, clearly presenting the concepts of Knowledge Graph Enhanced Learning and Knowledge Graph Augmented Inference. It effectively demonstrates the novel techniques applied in both the training and testing phases.

3. The performance across multiple benchmarks underscores the effectiveness of the approach, achieving performance gains of approximately 2-4%.

4. The ablation study, along with the time complexity analysis, highlights a favorable trade-off between accuracy and computational cost.

Weaknesses:

1. The motivation and explanations in some sections could be further clarified and strengthened (see Questions 1-3 for specific examples).

2. A discussion on how the proposed methods can be extended to other relevant scenarios as GMM models continue to develop would significantly enhance the paper's impact.

3. Including some failure cases would provide a more comprehensive evaluation of the method and give readers a better understanding of its limitations.

---

> ### Author Rebuttal · Authors · 2025-07-30
>
> Thank you for your valuable comments. We address your questions or concerns below.
>
> > **W1.1: About the motivation behind the "Trim to Key Relationships" step.**
>
> The motivation behind the "Trim to Key Relationships" step is twofold. First, it aims to retain only the most essential relationships so that the model can identify specific categories without generating excessively long outputs. Second, it focuses on preserving the most discriminative relationships to avoid different categories sharing the same relationships, which could lead to ambiguity.
>
> > **W1.2: Why does "GMM + Description Labels" not perform well.**
>
> The original GMM approach uses CLIP to match the text output of the MLLM model, but CLIP performs suboptimally with longer text inputs like description labels, as well as with retrieval tasks within the same modality [A]. In contrast, GMM combined with a knowledge graph benefits from more structured and concise attribute outputs, which allows for more accurate localization using the knowledge graph itself.
>
> [A] Cross the Gap: Exposing the Intra-modal Misalignment in CLIP via Modality Inversion
>
> > **W1.3: Clarify the meaning of  Fig. 5.**
>
> The green and blue lines correspond to accuracy and should be read against the left y-axis, while the orange line represents inference time and should be read against the right y-axis. As the maximum number of tokens increases, the generated descriptions become more detailed, leading to higher classification accuracy. However, the time required to generate a certain number of tokens is relatively fixed per token. Therefore, both the green and blue lines share the same orange line for inference time. We will improve the clarity of Fig.5 in the revised version.
>
> > **W2: About the discussion on how the method generalizes to broader scenarios.**
>
> To verify the adaptability of our method across broader scenarios, we conducted experiments on several fine-grained and medical imaging datasets. The specific experimental settings are as follows:
>
> - CUB-200 is a fine-grained bird species dataset with detailed part-level attributes. We construct a knowledge graph using each class's most certain attributes (confidence score 4), such as has_bill_shape::hooked or has_bill_shape::cone, resulting in approximately 17 relations per class after our graph construction method. All 200 classes are equally separated into 10 tasks.
> - HAM-10000 is a dermatoscopic image dataset of 10,000 pigmented skin lesions of 7 classes for melanoma classification. Since we could not find a suitable knowledge graph for medical data, we instead used DeepSeek-VL-O1 to generate 10 representative attributes for each of the 7 classes (such as color_dominance::light brown, background_skin::No pigmentation), serving as their semantic descriptions or attribute sets. We follow [B] to separate 7 classes in 3 tasks, containing [2,2,3] classes each.
>
> [B] BiasPruner: Debiased Continual Learning for Medical Image Classification, MICCAI 2024
>
> | datasets | cub200 |      | HAM10000 |      |
> |:--------:|:------:|:----:|:--------:|:----:|
> |          |   Avg  | Last |    Avg   | Last |
> |    GMM   | 49.34       | 40.39     |  79.22        |  63.81    |
> |  KG-GMM  |  52.65      |  43.84    |  81.45        |   64.34   |
>
> As shown in our experimental results, when relevant and visually discernible attributes (e.g., color, shape, pattern) are available, our model yields noticeable improvements over GMM, demonstrating that our approach is adaptable and benefits from contextual knowledge beyond ConceptNet.
>
> > **W3: About failure cases analysis.**
>
> We have analyzed several representative failure cases, which will be added to the revised manuscript. These cases reveal that while the integration of the knowledge graph helps the model attend to factual visual cues (e.g., red, AtLocation garage, RelatedTo barn), it can also introduce semantically misleading associations.
>
> For example, the model may misclassify fire_engine as pickup_truck due to shared contextual relations (e.g., both being AtLocation garage), or confuse fox_squirrel with red_fox due to the shared visual feature red. These errors suggest that while the relations are factually correct, they may not always align with the target classification semantics, especially when visual similarity overlaps with noisy or broad knowledge links. We are incorporating this analysis into the revised version to better contextualize the limitations and potential improvements for future work.
>
>
> | class ID | class name | model output | prediction |
> |:--------:|:----------:|------------|:----------:|
> |   n03345487        |  fire_engine               |  fire_engine. fire_engine AtLocation garage. fire_engine RelatedTo truck. fire_engine IsA machine.            |   pickup_truck         |
> |  n02088094         |   afghan_hound         |     horse. horse RelatedTo barn. horse Synonym horse. horse AtLocation farm_yard         |  barn          |
> |  n02356798         |   fox_squirrel             |  red_squirrel. red_squirrel DerivedFrom red. red_squirrel RelatedTo arctic_fox. red_squirrel IsA fox            |     red_fox       |
> |  n02363005         |   beaver                     |   fox. fox IsA animal. fox HasContext movies. fox RelateTo cartoons           |     red_fox       |

---

### Official Review · Reviewer_sh7m · 2025-06-27

**Clarity:** 3
**Significance:** 3
**Originality:** 3
**Rating:** 4
**Confidence:** 5

**Summary:**

To mitigate catastrophic forgetting in continual learning, this paper constructs an evolving knowledge graph to leverage relational information between classes, using KG-based labels during training and a KG-augmented strategy to refine predictions at inference. Experiments on benchmark datasets demonstrate strong performance in both conventional and few-shot continual learning settings, outperforming state-of-the-art methods.

**Questions:**

This paper introduces multi-modal models for continual learning, and existing multi-modal model based continual learning studies are suggested to be mentioned.

[1] Modalprompt: Dual-modality guided prompt for continual learning of large multimodal models.

[2] MLLM-CL: Continual Learning for Multimodal Large Language Models.

[3] HiDe-LLaVA: Hierarchical decoupling for continual instruction tuning of multimodal large language model.

**Ethical Concerns:**

["NO or VERY MINOR ethics concerns only"]

**Final Justification:**

The author's revisions successfully addressed my concerns. I maintain my positive score.

**Limitations:**

yes

**Quality:**

3

**Strengths And Weaknesses:**

Strengths:

- By incorporating knowledge graphs, the model captures semantic relationships between classes, enabling more discriminative feature learning for similar categories in incremental tasks.

- The KG-augmented inference strategy uses relational queries to guide model predictions, reducing misclassifications by leveraging prior knowledge. This effectively addresses the limitation of generative models drifting to higher-level categories.

- Experiments on diverse datasets and CIL settings (both conventional and few-shot CIL) validate the effectiveness of the proposed method.

Weaknesses:

- The KG trimming strategy (e.g., selecting top-two relations or two-hop connections) introduces subjectivity, potentially omitting discriminative relations critical for fine-grained class distinctions. Moreover, the approach relies heavily on ConceptNet, which may lack task-specific semantic relations in specialized domains (e.g., medical imaging or robotic vision), limiting generalizability to niche applications.
- While the study demonstrates KG-GMM’s superiority over GMM in exemplar-free settings, it does not evaluate whether the method maintains this advantage when competing with GMM augmented by data replay. Whether the KG’s benefits can surpass those of data replay, which can be validated by additional experiments.

---

> ### Author Rebuttal · Authors · 2025-07-30
>
> Thank you for your valuable comments. We address your questions or concerns below.
> > **W1.1: About KG trimming strategy in fine-grained classes.**
>
> In fine-grained classification, while distinguishing between classes may require more nuanced relations, including too many or overly complex ones can overwhelm the LLM, increase the likelihood of hallucinations, and degrade inference efficiency. Therefore, although slightly more relations may be necessary to capture subtle distinctions, careful trimming remains essential. It ensures that only the most informative and discriminative knowledge is retained, maintaining a balance between precision, model robustness, and computational efficiency.
>
> > **W1.2: About adaptability in specialized domains.**
>
> We choose ConceptNet because it integrates multiple structured knowledge sources (e.g., WordNet, DBPedia, OpenCyc) and offers rich, diverse, and readily accessible commonsense knowledge. It covers a wide range of entities and relations relevant to standard continual learning benchmarks, making it a practical and effective choice.
>
> To address scenarios where ConceptNet may not be applicable, we extend our approach in two ways: (1) For datasets with rich attribute annotations (e.g., CUB-200), we construct custom knowledge graphs based on these attributes. (2) For datasets lacking such annotations (e.g., HAM-10000), we utilize large pretrained models to generate visually grounded concepts and relations to form dataset-specific graphs. We include the corresponding setups and results to demonstrate the flexibility and generalizability of our method beyond ConceptNet.
>
> - CUB-200 is a fine-grained bird species dataset with detailed part-level attributes. We construct a knowledge graph using each class's most certain attributes (confidence score 4), such as has_bill_shape::hooked or has_bill_shape::cone, resulting in approximately 17 relations per class after our graph construction method. All 200 classes are equally separated into 10 tasks.
> - HAM-10000 is a dermatoscopic image dataset of 10,000 pigmented skin lesions of 7 classes for melanoma classification. Since we could not find a suitable knowledge graph for medical data, we instead used DeepSeek-VL-O1 to generate 10 representative attributes for each of the 7 classes (such as color_dominance::light brown, background_skin::No pigmentation), serving as their semantic descriptions or attribute sets. We follow [A] to separate 7 classes in 3 tasks, containing [2,2,3] classes each.
>
> [A] BiasPruner: Debiased Continual Learning for Medical Image Classification, MICCAI 2024
>
> | datasets | CUB-200 |      | HAM-10000 |      |
> |:--------:|:------:|:----:|:--------:|:----:|
> |          |   Avg  | Last |    Avg   | Last |
> |    GMM   | 49.34       | 40.39     |  79.22        |  63.81    |
> |  KG-GMM  |  52.65      |  43.84    |  81.45        |   64.34   |
>
> As shown in our experimental results, when relevant and visually discernible attributes (e.g., color, shape, pattern) are available, our model yields noticeable improvements over GMM, demonstrating that our approach is adaptable and benefits from contextual knowledge beyond ConceptNet.
>
> > **W2: About comparison with GMM augmented by data replay.**
>
>
> As shown below, our method outperforms GMM augmented by data replay on Tiny-ImageNet under both 100-5 and 100-10 task settings. Moreover, our method also achieves consistent performance improvements when augmented with exemplars.
>
> |        | exemplar | Tiny-ImageNet | Tiny-ImageNet     |   Tiny-ImageNet             |    Tiny-ImageNet  |    Tiny-ImageNet           |   Tiny-ImageNet   | ImageNet-R  |
> |--------|:-----:|:---------------:|:------:|:----------------:|:------:|:---------------:|:------:|:-------------:|
> |        |     | B100-5 tasks  |  B100-5 tasks    | B100-10 tasks | B100-10 tasks     | B100-20 tasks | B100-20 tasks     | B0-10 tasks |
> |        |     | Avg           | Last | Avg            | Last | avg           | Last | Last        |
> | GMM    |  &cross;   | 83.42              | 76.98     | 82.49               | 76.51     |  81.70             | 76.03     |  80.72           |
> | GMM    |  &check;   | 84.16              | 78.46     |  83.95              | 78.64      |  84.23             | 79.17     | 89.41            |
> | KG-GMM | &cross;    |  86.17             | 81.86     | 84.37               | 78.16     | 83.18              | 78.32     |  84.29           |
> | KG-GMM | &check;    |  **87.54**             | **82.35**     | **85.12**               | **79.32**     |  **85.17**            | **80.34**     | **90.11**             |
>
>
> > **Q1: About incorporating existing studies.**
>
> We will carefully revise the manuscript and incorporate the relevant literature into our work.

---

### Official Review · Reviewer_SjoA · 2025-07-03

**Clarity:** 3
**Significance:** 3
**Originality:** 3
**Rating:** 5
**Confidence:** 4

**Summary:**

This paper introduces KG-GMM, a novel knowledge graph-enhanced generative multi-modal framework designed to alleviate catastrophic forgetting in class-incremental learning (CIL). By incrementally constructing a task-aware commonsense knowledge graph, the model uses structured relational information to enrich label representation and guide relation-aware reasoning during inference. Extensive experiments on conventional and few-shot CIL benchmarks show that KG-GMM consistently outperforms previous methods.

**Questions:**

1. Consider exploring the impact of the choice of relation types (e.g., only IsA, AtLocation) on performance.

2. It is necessary to emphasize how relations of second hop supplement information when relation of one hop is not enough.

3. Similar classes in different tasks may influence the results. Thus, it would be helpful to run experiments on different seeds on TinyImageNet and ImageNet-R.

**Ethical Concerns:**

["NO or VERY MINOR ethics concerns only"]

**Final Justification:**

The responses address my concerns. Thus the final rating was upgraded.

**Limitations:**

The paper doesn’t include a discussion about the limitations. The suggestions are provided in Questions.

**Quality:**

3

**Strengths And Weaknesses:**

Strengths:

1. The idea of combining generative multi-modal LLMs with a dynamically growing knowledge graph for CIL is innovative. It addresses a critical limitation of prior generative CIL methods: the tendency to produce generic or biased predictions due to loss of class-specific detail.

2. The paper introduces a well-structured knowledge graph construction mechanism that selects informative relations per class and guides both learning and inference.

Weaknesses:

1. The approach heavily relies on ConceptNet, a commonsense KG. The proposed method is only applicable to datasets with category labels, like Domainnet.

2. The paper could benefit from a deeper qualitative analysis of failure cases. Some examples are shown in Fig. 4, but more insights into when the method fails to disambiguate similar classes (e.g., due to misleading relations) would be valuable.

---

> ### Author Rebuttal · Authors · 2025-07-29
>
> Thank you for your valuable comments. We address your questions or concerns below.
>
> > **W1: Regarding reliance on ConceptNet and category labels.**
>
> We choose ConceptNet because it integrates multiple structured knowledge sources (e.g., WordNet, DBPedia, OpenCyc) and offers rich, diverse, and readily accessible commonsense knowledge. It covers a wide range of entities and relations relevant to standard continual learning benchmarks, making it a practical and effective choice.
>
> To address scenarios where ConceptNet may not be applicable, we extend our approach in two ways: (1) For datasets with rich attribute annotations (e.g., CUB-200), we construct custom knowledge graphs based on these attributes. (2) For datasets lacking such annotations (e.g., HAM-10000), we utilize large pretrained models to generate visually grounded concepts and relations to form dataset-specific graphs. We include the corresponding setups and results to demonstrate the flexibility and generalizability of our method beyond ConceptNet.
>
> - CUB-200 is a fine-grained bird species dataset with detailed part-level attributes. We construct a knowledge graph using each class's most certain attributes (confidence score 4), such as has_bill_shape::hooked or has_bill_shape::cone, resulting in approximately 17 relations per class after our graph construction method. All 200 classes are equally separated into 10 tasks.
> - HAM-10000 is a dermatoscopic image dataset of 10,000 pigmented skin lesions of 7 classes for melanoma classification. Since we could not find a suitable knowledge graph for medical data, we instead used DeepSeek-VL-O1 to generate 10 representative attributes for each of the 7 classes (such as color_dominance::light brown, background_skin::No pigmentation), serving as their semantic descriptions or attribute sets. We follow [A] to separate 7 classes in 3 tasks, containing [2,2,3] classes each.
>
> [A] BiasPruner: Debiased Continual Learning for Medical Image Classification, MICCAI 2024
>
> | datasets | CUB-200 |      | HAM-10000 |      |
> |:--------:|:------:|:----:|:--------:|:----:|
> |          |   Avg  | Last |    Avg   | Last |
> |    GMM   | 49.34       | 40.39     |  79.22        |  63.81    |
> |  KG-GMM  |  52.65      |  43.84    |  81.45        |   64.34   |
>
> As shown in our experimental results, when relevant and visually discernible attributes (e.g., color, shape, pattern) are available, our model yields noticeable improvements over GMM, demonstrating that our approach is adaptable and benefits from contextual knowledge beyond ConceptNet.
>
> > **W2: More qualitative analysis of failure cases.**
>
> We have analyzed several representative failure cases, which will be added to the revised manuscript. These cases reveal that while the integration of the knowledge graph helps the model attend to factual visual cues (e.g., red, AtLocation garage, RelatedTo barn), it can also introduce semantically misleading associations.
>
> For example, the model may misclassify fire_engine as pickup_truck due to shared contextual relations (e.g., both being AtLocation garage), or confuse fox_squirrel with red_fox due to the shared visual feature red. These errors suggest that while the relations are factually correct, they may not always align with the target classification semantics, especially when visual similarity overlaps with noisy or broad knowledge links. We are incorporating this analysis into the revised version to better contextualize the limitations and potential improvements for future work.
>
> | class ID | class name | model output | prediction |
> |:--------:|:----------:|------------|:----------:|
> |   n03345487        |  fire_engine               |  fire_engine. fire_engine AtLocation garage. fire_engine RelatedTo truck. fire_engine IsA machine.            |   pickup_truck         |
> |  n02088094         |   afghan_hound         |     horse. horse RelatedTo barn. horse Synonym horse. horse AtLocation farm_yard         |  barn          |
> |  n02356798         |   fox_squirrel             |  red_squirrel. red_squirrel DerivedFrom red. red_squirrel RelatedTo arctic_fox. red_squirrel IsA fox            |     red_fox       |
> |  n02363005         |   beaver                     |   fox. fox IsA animal. fox HasContext movies. fox RelateTo cartoons           |     red_fox       |
>
> >**Q1: About the choice of relation types.**
>
> We compare using a fixed set of relations (only IsA, IsA + AtLocation, IsA + AtLocation + HasProperty) with our method that selects key relations adaptively. Our approach performs best. This is because rare classes, such as West Highland white terrier, often lack common relations like IsA or AtLocation, leaving only the class name and leading to information loss. Moreover, fixed relation sets tend to produce overlapping links, where multiple classes are associated with the same entities, increasing classification ambiguity. Therefore, selectively retaining specific relations without considering their availability per class leads to suboptimal results.
>
> | relations | IsA | IsA + AtLocation | IsA + AtLocation + HasProperty| Ours |
> |:--------:|:----------:|:------------:|:----------:|:----:|
> | Last Acc  | 81.23    | 81.75                 |   82.79     |   84.29   |
>
> > **Q2: On the importance of the second-hop relations.**
>
> We aim to select relations that best distinguish all seen classes. When certain relations have already been used by previous learned classes and the current class lacks available one-hop relations, we explore second-hop relations. This has two key benefits: it avoids reusing the same relations, which could increase classification ambiguity, and it encourages the model to focus on distinctive features when differentiating similar classes, thus improving classification performance.
>
> To validate this, we conducted comparative experiments across three settings:
>
> - One-hop relations (fallback to class name if no relation),
>
> - Class name only (GMM), and
>
> - Our adaptive method with second-hop relations.
>
> | Method| one hop |   class names(GMM) | Ours     |
> |:--------:|:------:|:--------:|:----:|
> |    Last Acc| 83.68       |   80.72        |  84.29    |
>
> Our method yields a 0.61 absolute improvement over the one-hop setting using the same number of relations, confirming the added value of second-hop relations in enhancing model discrimination and overall performance.
>
> >**Q3: Regarding different class orders.**
>
> We agree that in our method, the class order may affect the continual learning performance; thus, we conducted experiments on different seed-based class orders in Appendix Sec A.1. The results show that despite slight variations in results across different orderings, our model maintains stable improvements over other baselines.

---

### Comment · Area_Chair_LoZQ · 2025-08-04
**Please check authors' rebuttal**

Dear reviewers,

Thank you for your efforts. The authors have provided a detailed response. Please check if any concerns remain and engage in discussion with authors.

Best,
AC

---

### Note · Authors · 2025-08-12

Dear AC and Reviewers:

We sincerely thank all reviewers for their time and valuable feedback. We appreciate their recognition of our method as novel and meaningful (Reviewer SjoA and ErR5), our paper is well written (Reviewer qKfb) and performance superior (Reviewer sh7m and ErR5).

Overall, we have comprehensively addressed all reviewer comments and concerns point by point. Below is a brief summary.

## **Regarding reliance on ConceptNet and adaptability in specialized domains**

In our response, we have supplemented the experimental results of our method on fine-grained datasets (Aircraft, CUB-20, Herbarium 19) and the medical image dataset (HAM-10000). Our approach consistently improves upon the GMM baseline provided that relevant feature data is available to construct a graph (e.g., attribute information in CUB-200, or lesion characteristics like color/shape in HAM-10000). This demonstrates the broad applicability of our method.

## **More qualitative analysis of failure cases**

We have provided detailed examples and corresponding analysis of our method's failure cases, demonstrating its characteristics and potential limitations.

## **Regarding the plasticity–stability dilemma in continual learning**

In our response to Reviewer qKfb, we described how traditional methods address the plasticity-stability issue, how GMM tackles this issue, and how our method further resolves it.

## **Other Detailed Comments**

We have provided responses to all other concerns including choice of relation types, second-hop relations, comparison with GMM augmented by data replay.

**Overall, we are trying to address all concerns raised by reviewers and polish our paper in the revised version to improve the completeness and clarity of our paper, specifically:**

1. We will incorporate experiments on fine-grained and medical image datasets in the revised version
2. We will include the failure cases of our method in Section 4.4 and Figure 4.
3. We will revise our introduction section to better express our motivation and add explanation on how our method addresses the plasticity and stability dilemma in continual learning.

Finally, we sincerely appreciate the time and effort invested by all reviewers and Area Chair in evaluating our manuscript. Thank you once again for your valuable contributions.

---

### Decision · Program_Chairs · 2025-09-17

**Decision:**

Accept (poster)

**Comment:**

The paper proposes to use knowledge graph generation to maintain the model's knowledge or previous tasks in the context of Incremental classification. The authors build on top MiniGpt4 and consider several tasks with no exemplars while only updating the linear projection layer.

Limitations mainly included the reliance on Concept Net for knowledge graph generation and concept classes, and other experimental limitations that the authors have addressed in the rebuttal.

All reviewers appreciated the contribution of the paper and the rebuttal has addressed their concerns.